# Monitoring and Inference of Behavioral Resistance in Beneficial Insects to Insecticides in Two Pest Control Systems: IPM and Organic

José Alfonso Gómez-Guzmán 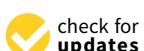, María Sainz-Pérez and Ramón González-Ruiz *

Department of Animal Biology, Vegetable Biology and Ecology, University of Jaén, 2307 Jaén, Spain; jgguzman@ujaen.es (J.A.G.-G.); mariasainzperez@gmail.com (M.S.-P.)
* Correspondence: ramonglz@ujaen.es; Tel.: +34-953-212-499

**Abstract:** Pyrethrins are the most widely used insecticide class in olive groves with organic management. Although there are data sets about insect pests of stored products and human parasites developing resistance to pyrethrins, there is no information on the long-term effect on olive agroecosystems. A field method based on the experimental induction of sublethal effects by means of insecticide application, and the monitoring of the response of insects through post-treatment sampling, has recently been developed. This method has allowed for the detection of populations behaviorally resistant to organophosphates in integrated pest management (IPM) and conventional crops. With the application of a similar methodology, this study aimed to verify the possible reaction of natural enemies in organic crops, using pyrethrins as an inducing insecticide. The study was carried out in 2019 in two olive groves in southern Spain (Jaén, Andalusia), one of them being IPM and the other being an organic production system. The results did not allow for verification of the behavioral resistance in populations of natural enemies of both IPM and organic management against pyrethrins, while against dimethoate, behavioral resistance was verified in IPM management. The possible causes involved in obtaining these results are discussed.

**Keywords:** behavioral resistance; beneficial insects; dimethoate; integrated pest management (IPM); olive growing; organic pest management; pyrethrins



## 1. Introduction

After being applied to crops, insecticides are subjected to environmental factors (light, heat, water, and wind), which cause gradual degradation and dispersion of their residues, which implies that affected insects are exposed to sublethal doses over time [1–4]. A sublethal dose/concentration is defined as inducing no apparent mortality in the experimental population [4,5]. In general, insecticide dose/concentrations under the median lethal ($LD_{50}/LC_{50}$) are considered to induce sublethal effects [5]. The induced effects produce alterations that affect physiological, biological, and behavioral processes, among which agitation, hyperreflexia, irritability, and repellency have been reported [3–9]. These effects allow individuals affected by doses lower than the $LD_{50}$ to avoid new contact with insecticide residues and to escape their toxic action.

The repellent effect of nonselective insecticides is a characteristic reported for organochlorines [10,11], pyrethroids [11–15], organophosphates [11,16], and carbamates [11]. Although the precise behavioral responses of insects in the field are elusive and difficult to measure [17], experiments carried out in olive groves by monitoring populations of natural enemies (*Aeolothrips intermedius* (Bagnall, 1934) (Thysanoptera: Aeolothripidae); *Chrysoperla agilis* (Henry et al., 2003) (Neuroptera: Chrysopidae); *Harraphidia laufferi* (Navás, 1915) (Raphidioptera, Raphidiidae); *Anthocoris nemoralis* (Fabricius, 1794) (Hemiptera, Anthocoridae); *Orius laevigatus* (Fieber, 1860) (Hemiptera, Anthocoridae)) after insecticide applications have observed repellency in crops submitted to frequent dimethoate application [18,19], which has been used in conventional pest

management and IPM in recent decades [20,21]. The applied method is based on the monitoring of insect populations by means of the deployment of adherent traps, installed immediately after insecticide application. In the results, negative deviations have been observed in the capture rates of beneficial insects in traps of organic crops after experimental insecticide application. On the contrary, positive deviation is found in agrosystem populations regularly submitted to synthetic insecticide applications, which is indicative of the repellency reaction exhibited by behaviorally resistant lineages. This repellent effect, manifested by insect populations after repeated synthetic insecticide applications, has been interpreted as the first barrier of a detoxification mechanism [9,12,22–25]. As frequently reported [5,9,26], in these agrosystems, the environmental pressure represented by the regular use of insecticides has led to a gradual and progressive purification of behaviorally resistant lineages.

Despite the decisive effect of sublethal doses [3], their effect has been underestimated [27,28] and the need for more precise evaluations of their impact has been highlighted [4,29]. Their importance acquires greater relevance in the characterization of organic crops, compared to those based on the use of synthetic insecticides. The importance of organic cultivation in recent years [30,31] is reflected in the 5% average annual increase in the cultivated area, currently representing 10% of the total olive grove cultivation area in Spain [32]. Compared to traditional agriculture, the higher quality of products of organic origin leads to greater respect for the environment, as well as the progressive elimination of synthetic chemical products. Regarding pest control, biodiversity stimulation in organic crops will have an impact on the improvement of the entomophagous activity of natural enemies, which has been verified in the case of *Prays oleae* (Bernard, 1788) (Lepidoptera: Praydidae) [33].

In order to promote conversion to organic agricultural management regimes among farmers, the regional government grants economic subsidies. In addition to these benefits, higher prices for organic olive oil have been set in the market, which have increased by 15–30% [34]. In recent years, the granting of economic subsidies by the regional government has triggered a 40% increase in the area of organic olive groves in relation to the year 2001 [35]. Among the requirements demanded by the Regional Administration to grant the certification of organic olive groves, crops with this type of management regime must undergo periodic inspections, which allow the determination of pesticide residues in olive oil. However, alternative and/or complementary procedures are required. In this sense, the absence of behavioral resistance in the beneficial insects in relation to synthetic insecticides is a distinctive characteristic of organic pest management with respect to conventional management and IPM [33]. This has been the determining element for the development of a reliable strategy for its characterization from a behavioral point of view. This method is based on the experimental application of an organophosphate insecticide in a reduced sector of problem olive groves, where changes in capture rates are subsequently monitored, during the following two weeks, for which adherent chromotropic traps are used. Although its relative abundance is lower in olive groves usually treated with synthetic insecticides, the capture rates in traps are higher after treatment application. This increase in capture rates is attributed to the repellency reaction generated by insecticide application, which occurs when insect populations in agrosystems are frequently exposed to synthetic insecticides. This fact constitutes an essential difference with respect to organic olive groves, in which the relative abundance is higher and the post-treatment capture rates of beneficial insects are lower, as a consequence of the absence of behaviorally resistant lineages.

Among organic alternatives to conventional synthetic insecticides, natural pyrethrins, obtained from Dalmatian pyrethrum (*Tanacetum cinerariifolium*), have important advantages such as their low toxicity for mammals [36–38] and short environmental persistence [39] due to the action of UV rays, so their effect on agroecosystems is of very short duration [40,41]. The lower persistence and toxicity to humans make pyrethrins more ecologically acceptable than the widely used conventional agricultural synthetic insecticides [42]. Among the sublethal effects of pyrethrins, the reduction in longevity and fecundity in certain parasitoid hymenopterans [43] and their repellent action on various Diptera species, such as

*Aedes aegypti* (Linnaeus, 1762) (Diptera: Culicidae) or *Aedes albopictus* Skuse, 1895 (Diptera: Culicidae) [44–46], have been frequently reported. Most data on the resistance to pyrethrins come from programs for protection against dipteran stings [44–52], as well as conservation programs for stored agricultural products [53–57]. In the latter, cross-resistance against organophosphates and pyrethrins has been detected [58], which suggests the hypothesis that a similar effect may be occurring in olive growing given the frequent behavioral resistance to dimethoate [59,60]. With this study, the application of the monitoring induction technique is proposed, which has been very useful in detecting lineages resistant to synthetic insecticides, to detect possible behavioral resistance of insects to pyrethrins, as it is the most commonly applied insecticide in organic olive growing.

## 2. Materials and Methods

### 2.1. Description of the Study Area

This study was carried out in two olive groves in the municipality of Jaén (Andalusia, southern Spain) in the spring of 2019, during the oviposition period of the phyllophagous generation of *P. oleae* in olive flowers, a time of great activity of the auxiliary entomofauna [61,62]. The selected olive groves (Figure 1) correspond to two different types of pest management: IPM and organic management (ORG).

The integrated pest management (IPM) olive grove has an area of 20 ha (37°52′12.55″ N, 3°34′03.33″ W). Since 1995, the management of this olive grove has been carried out in accordance with the principles of integrated pest management. The 28-year-old olive trees belong to the Picual variety and are arranged in a 10 × 10 m configuration. The soil is fertilized with organic and mineral nutrients twice a year. In addition, foliar fertilization is annually carried out using crystalline urea (nitrogen content of 46%), potassium sulfate, and natural amino acids (arginine, glycine, threonine, and proline). Pest control is only carried out when the population levels of pest species exceed the action threshold of damage established in integrated production regulations. When this occurs, the insecticides used are Dimethoate 40% © (IRAC Group 1B) for the control of the olive fly *Bactrocera oleae* (Rossi, 1790) (Diptera: Tephritidae), the olive bark beetle *Phloeotribus scarabaeoides* (Bernard, 1788) (Coleoptera: Curculionidae), and the olive moth *P. oleae*, and chlorpyrifos 48% © (IRAC Group 1B) for the control of the branch borer *Euzophera pinguis* (Haworth, 1811) (Lepidoptera: Pyralidae) (Table 1).

**Table 1.** Products applied in the control of the main pests and the annual application regime in both olive groves (IPM and organic).

| | | *Euzophera pinguis* | *Prays oleae* | | *Phloeotribus scarabaeoides* | *Bactrocera oleae* | |
|---|---|---|---|---|---|---|---|
| **IPM** | Pesticide: active principles | Pyrinex®: Chlorpyrifos 48% | Danadim®: Dimethoate 40% | | Danadim®: Dimethoate 40% | Danadim®: Dimethoate 40% | |
| | Active ingredient per ha | 0.3 L/ha | 0.3 L/ha | | 0.45 L/ha | 0.9 L/ha | |
| | No. applications (date) | 1 per year (May) | 1 per year (May) | | 1 per year (June) | 2 per year (September + October) | |
| **ORG** | Device/Pesticide: active principles | Delta traps + pheromone dispenser | Abanto®: Pyrethrin 4% | *B. thuringiensis aizawaii* | trap logs | Abanto®: Pyrethrin 4% | Spintor®: Spinosad 0.024% |
| | Active ingredient per ha | 1 trap/ha | 0.6 L/ha | 1–2 kg/ha | - | 0.6 L/ha | 1 L/ha |
| | No. applications (date) | 1 per year (May) | 1 out of 4 years (May) | 3 out of 4 years (May) | - | 2 out of 5 years (September + October) | 1/year (October) |

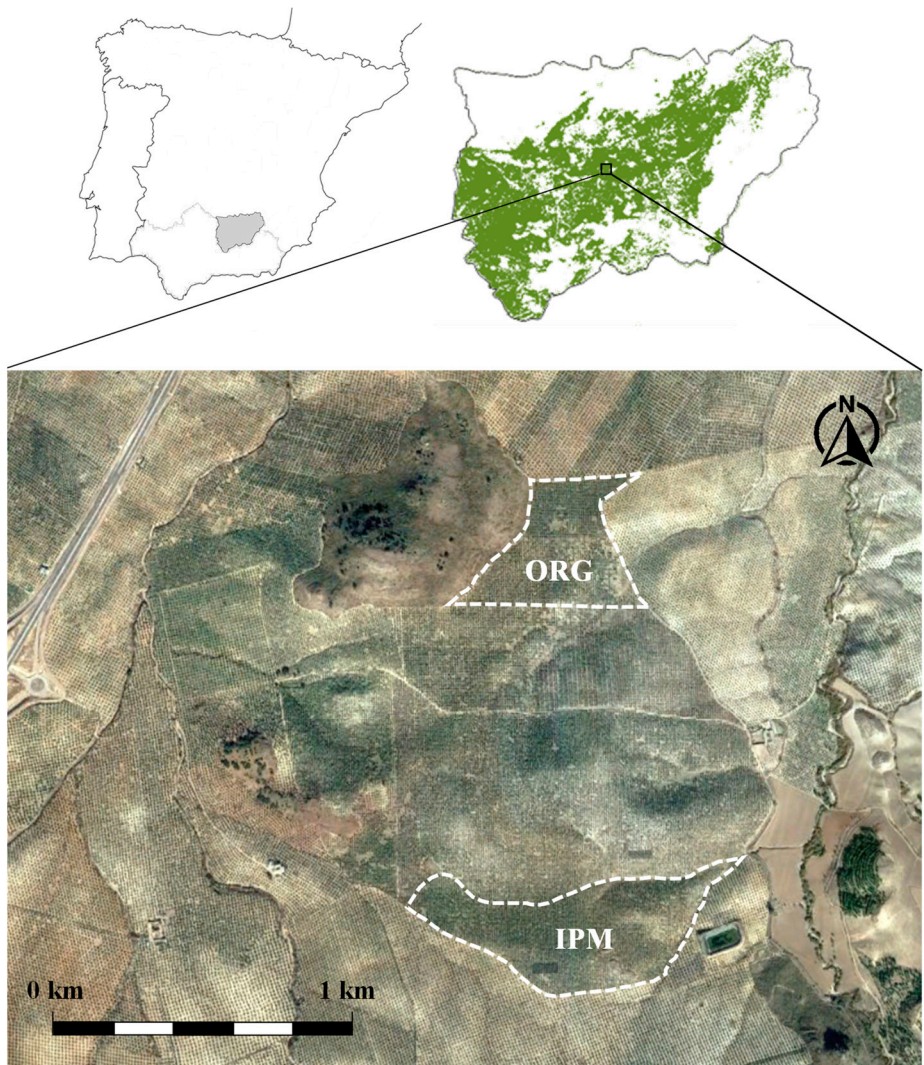

**Figure 1.** Location of the selected olive groves in the municipality of Jaén: Integrated pest management (IPM) and organic (ORG). Source: Own elaboration using the Google Earth Pro geographic information system.

The organic (ORG) olive grove has an area of 17 ha (37°52′21.11″ N, 3°34′29.46″ W). This olive grove has had organic certification since 2007. The olive trees, of the Picual variety, are 28 years old and are arranged in a 10 × 10 m configuration. The soil is biannually fertilized with natural organic nutrients, and foliar treatments are annually carried out with biostimulants authorized for use in organic farming (CE Regulation No. 889/2008). The insecticides used for pest control are of natural origin, authorized in Annex II of the EC Regulation No. 889/2008 (Table 1). For the control of *B. oleae*, the annual patching technique is used, and 0.024% spinosad (IRAC Group 5) is applied in combination with a protein hydrolysate. Additionally, and in order to manage resistance to spinosad [63–67], in two out of every five years, this treatment is complemented with the application of 4% pyrethrins (IRAC Group 11A). For the control of *P. oleae*, homogeneous spray application of a *Bacillus thuringiensis* var. *aizawaii* (IRAC Group 3A) formulation is performed, with a frequency of three out of every four years. Furthermore, in years with the occurrence of strong attack, which occurs at an average frequency of one out of every four years [61], this application is replaced by a homogeneous spray of 4% natural pyrethrins. The control of *E. pinguis* is based on a massive trapping technique using delta traps baited with synthetic pheromone [68]. Finally, to control *P. scarabaeoides*, bait trunks are used during the period of attack on pruning logs. These trunks are destroyed before the emergence of the offspring.

It is noteworthy that the two olive groves are cultivated in rainfed systems and that, in both, the growth of a spontaneous vegetation cover is encouraged, controlled with mechanical mowing at specific times. For this reason, a well-preserved cover has developed in both olive groves, where the majority of species are *Senecio vulgaris* Linnaeus, *Diplotaxis virgata* Candolle, *Bromus madritensis* Linnaeus, *Lolium rigidum* Gaudin, *Hordeum leporinum* Linnaeus, *Sinapis alba* Linnaeus, *Anacyclus clavatus* Desfontaines, and *Crepis sancta* Linnaeus. The olive groves of both crops are pruned every two years.

### 2.2. Experimental Design

Both olive groves (organic and IPM) were divided into three blocks, establishing in each of them three plots of 40 × 40 m (16 olive trees), and the minimum distance between plots of the same block was 200 m (Figure 2). Within each block, two of the three plots were randomly selected to apply the two insecticides under study. Thus, in each olive grove (organic and MIP), a plot of each block was sprayed with 40% Dimethoate © (400 g/L) (BASF) at a concentration of 0.1% (*v/v*); another plot of each block was sprayed with nonsystemic bioinsecticide Abanto © (4% pyrethrins), a natural compound allowed in organic agriculture (CE No. 889/2008); and finally, the third plot of each block was established as a control, for which distilled water was applied. These experimental treatments were carried out on 14 May under calm atmospheric conditions and with a wind speed of less than 5 km/h. Treatments were carried out using a MATABI Evolution 16 © hydraulic sprayer. The area treated with insecticides was equivalent to 4.8% and 5.6% of the total area of IPM and organic olive groves, respectively, being that the latter has maintained the organic cultivation certification. It should be noted that, during this work, in the rest of the two olive groves considered, no type of insecticide application was carried out since this could interfere with the results.

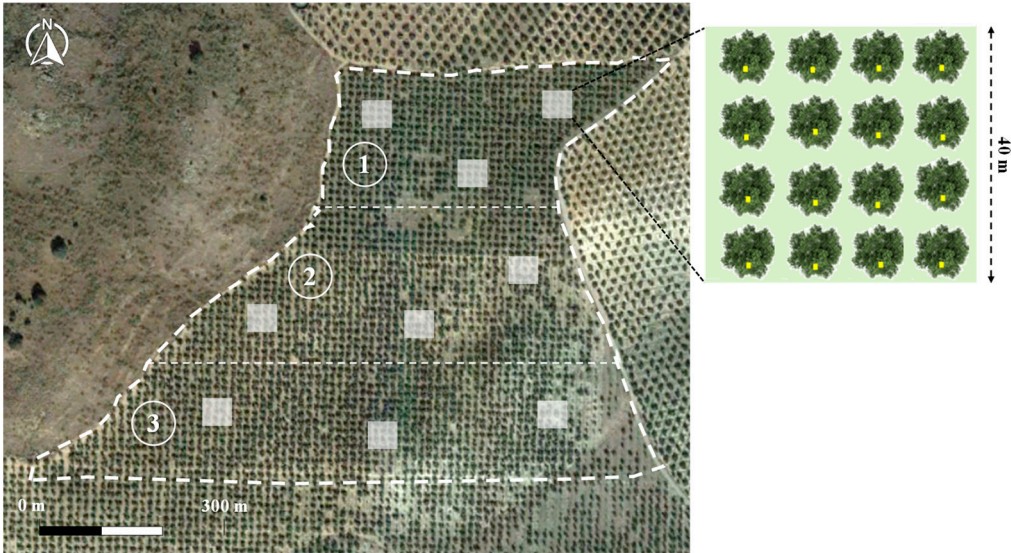

**Figure 2.** Distribution of the plots into the three organic olive grove blocks. Source: Own elaboration using the Google Earth Pro geographic information system.

In all plots of both olive groves, the main natural enemy species of olive tree pests were monitored. To do this, the passive sampling technique was carried out using adherent yellow chromotropic traps (20 × 40 cm). This sampling methodology has provided excellent results and stands out for being easily replicable [33,62,69–71]. Traps were immediately placed after experimental insecticide applications, at the rate of one per olive tree (16 traps per plot) and at a height of 1.5 m in the N sector of each tree. Since their placement, the traps were weekly renewed, establishing two sampling intervals: 14–21 May and 21–28 May. After their renewal, the traps were temporarily stored in a cold room (4 °C). In each sam-

pling interval and from each block of both olive groves, 6 traps of each plot were randomly selected to be observed. This subsample of traps from each block was used in order to avoid the lack of randomization of the observed samples and prevent pseudoreplication effects. Therefore, in each sampling interval the number of traps observed of each treatment (dimethoate, pyrethrins, and control) in each olive grove was 18. The traps were examined by means of a binocular magnifying glass for the taxonomic determination and quantification of beneficial species captured. For this, those species associated with at least one olive pest species [72–76] were taken into account.

To obtain an estimate of the relative abundance of the different species in both types of olive grove management systems, the capture values recorded in the control plots were considered, as they were free of any alteration caused by insecticide applications.

*2.3. Statistical Analysis*

For the statistical analysis of the data, the Statgraphics Centurion XVII statistical package (2016) has been used. The normality of the distributions was verified by the Shapiro–Wilk normality test. Since the data set does not fit to a normal distribution, the Mann–Whitney U test was applied to determine significant differences between the captured values obtained from the different natural enemy species in control plots of the two management regimes. To determine significant differences among the captured values obtained from the species of beneficial insects in plots of the different treatments (dimethoate, pyrethrins, and control), the Kruskal–Wallis test was used. Once statistically significant differences were determined using the Kruskal–Wallis test, the Mann–Whitney U test was used to compare the catch values of the natural enemies species between the pair of treatments.

## 3. Results

### 3.1. Analysis of the Relative Abundance of Beneficial Insects (Control Plots)

Among the captured species, 12 species of natural enemies were selected (Table 2), which are relatively common in the entomofauna associated with olive grove pests in southern Spain [33,72,77,78]. Individuals of these 12 species of natural enemies were captured in both olive groves. Due to its greater abundance, the *A. intermedius* predator stands out, which is a cosmopolitan species common in a wide range of crops [33,79,80]. Most of the captured species (nine species) are natural enemies of the olive moth *P. oleae*, which, as indicated, during sampling, is in the oviposition period corresponding to the anthophagous generation. Among parasitoids are the following hymenopterans predominate: *Tetrastichus cesirae* (Russo, 1938), *Elasmus flabellatus* (Fonscolombe, 1832) (Eulophidae), *Ageniaspis fuscicollis* (Dalman, 1820) (Encyrtidae), and *Diadegma semiclausum* (Hellen, 1949) (Ichneumonidae). Despite the predatory importance of chrysopids in the control of *P. oleae* [81–84], the species *Ch. agilis* (Henry et al., 2003) has presented exceptionally low values, most likely because its predatory activity takes place mainly during the oviposition period corresponding to the carpophagous generation, which occurs during the months of June and July [85].

The relative abundance of beneficial species in the olive groves of both types of management is represented in Figure 3 (control plots). All species considered presented relative abundance values much higher in the organic olive grove compared to the IPM olive grove. Figure 4 represents the results of the comparison test for the capture values of beneficial species with the highest relative abundance. In all of them, the existence of significant differences in favor of the organic olive grove stands out (*p* < 0.05).

**Table 2.** Identified natural enemy species; olive grove pests that prey/parasitize and bibliographical references.

| | Identified Species | Associated Olive Pest Species | Bibliographic References |
|---|---|---|---|
| **Predators** | *Anthocoris nemoralis* (F., 1794) (Hem.: Anthocoridae) | *Euphyllura olivina*; *Liothrips oleae*; *Prays oleae* | Arambourg, 1986; Andrés-Cantero, 1997; Bejarano-Alcázar et al., 2011 |
| | *Orius laevigatus* (Fieber, 1860) (Hem.: Anthocoridae) | *Euphyllura olivina*; *Liothrips oleae*; *Prays oleae* | Arambourg, 1986; Andrés-Cantero, 1997; Bejarano-Alcázar et al., 2011 |
| | *Chrysoperla agilis* Henry et al., 2003 (Neu.: Chrysopidae) | *Euphyllura olivina*; *Prays oleae* | Arambourg, 1986; Bozsik et al., 2009 |
| | *Harraphidia laufferi* (Navás, 1915) (Rap.: Raphidiidae) | *Phloeotribus scarabaeoides* | González-Ruiz, 1989; Rozas & González-Ruiz, 2017 |
| | *Aeolothrips intermedius* Bagnall, 1934 (Thy.: Aeolothripidae) | *Liothrips oleae*; *Aceria oleae*; *Oxycenus maxwelli* | De Liñán, 1998 |
| **Parasitoids** | *Chelonus elaeaphilus* Silvestri, 1908 (Hym.: Braconidae) | *Prays oleae* | Campos, 1981; Arambourg, 1986; Carrero, 1996 |
| | *Ageniaspis fuscicollis* (Dalman, 1820) (Hym.: Encyrtidae) | *Prays oleae* | Campos, 1981; Arambourg, 1986; Carrero, 1996 |
| | *Pnigalio mediterraneus* Ferrierre & Delucchi, 1957 (Hym.: Eulophidae) | *Bactrocera oleae*; *Prays oleae* | Neuenschwander et al., 1983; El-Heneidy et al., 2001 |
| | *Tetrastichus cesirae* Russo, 1938 (Hym.: Eulophidae) | *Saissetia oleae*; *Bactrocera oleae*; *Liothrips oleae* | Arambourg, 1986; De Andrés Cantero, 1997 |
| | *Elasmus flabellatus* (Fonscolombe, 1832) (Hym.: Eulophidae) | *Prays oleae* | Campos, 1981; Nave et al., 2017 |
| | *Diadegma semiclausum* (Hellén, 1949) (Hym.: Ichneumonidae) | *Prays oleae* | De Andrés Cantero, 1997 |
| | *Itoplectis alternans* (Gravenhorst, 1829) (Hym.: Ichneumonidae) | *Prays oleae* | De Andrés Cantero, 1997 |

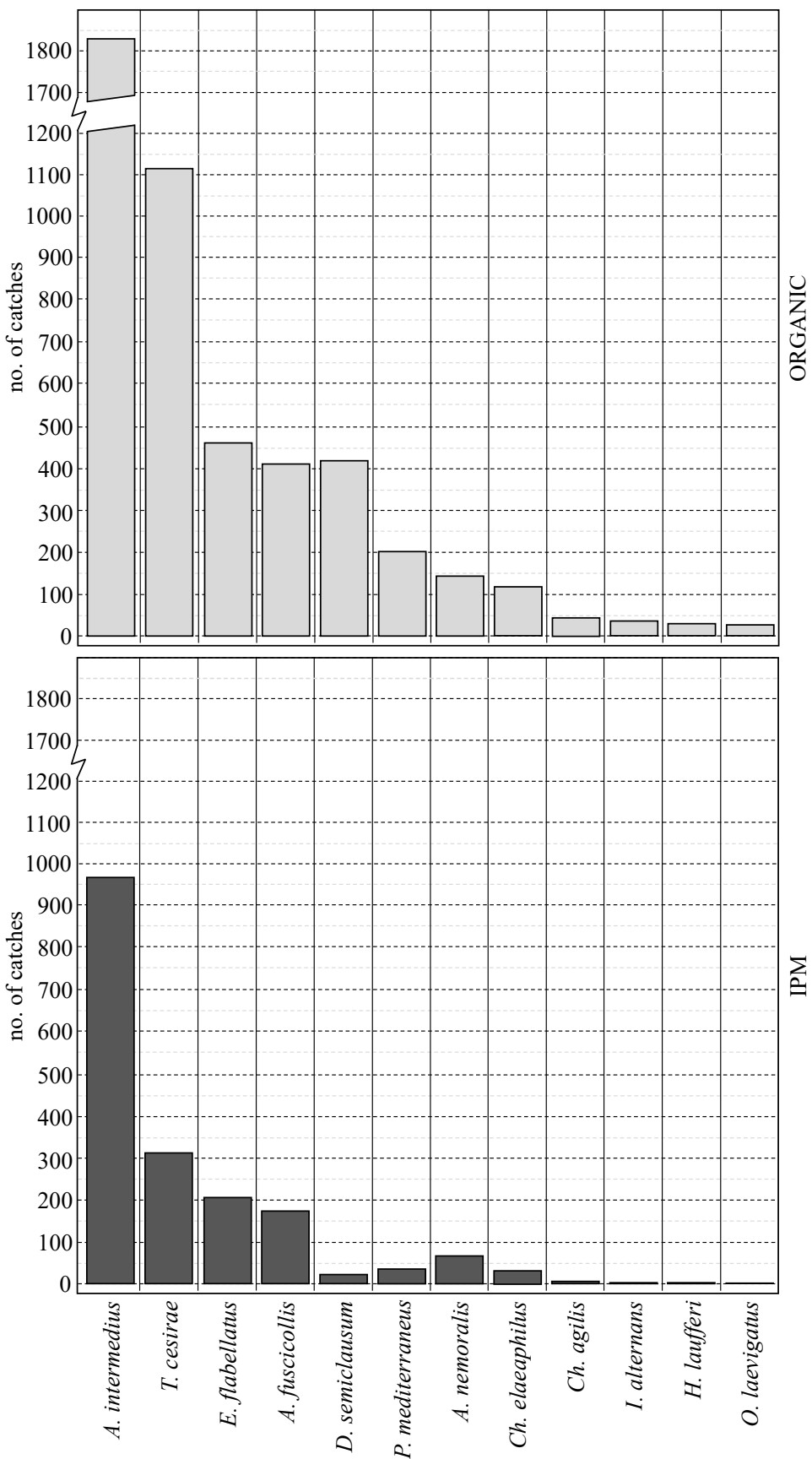

**Figure 3.** Abundance values obtained by the species in the control plots of the organic (light color) and IPM (dark color) olive groves.

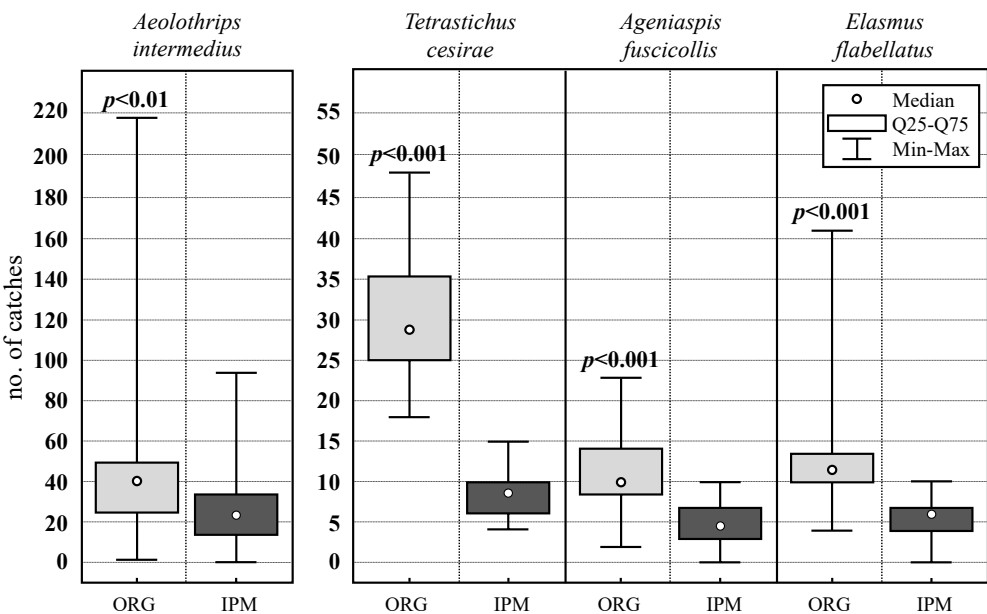

**Figure 4.** Statistic capture values (median, quartile 25, quartile 75, maximum, and minimum) of the major beneficial species in the control plots of organic (light color) and IPM (dark color) olive groves. The level of statistical significance obtained from the Mann–Whitney U test is also indicated (*p*-value).

### 3.2. Behavioral Resistance Assessment (Plots Treated with Dimethoate/Pyrethrins)

Figure 5 shows the statistic capture values of the main species in the control plots and the plots treated with both types of insecticides. The results of the Kruskal–Wallis test allowed rejecting the null hypothesis of equality of medians of capturing data in the three experimental treatments of both olive groves: *A. intermedius* (ORG: Chi-square = 16.21; df = 2; $p < 0.001$; IPM: Chi-square = 14.22; df = 2; $p < 0.001$), *T. cesirae* (ORG: Chi-square = 66.16; df = 2; $p < 0.001$; df = 2; IPM: Chi-square = 72.79; df = 2; $p < 0.001$), *A. fuscicollis* (ORG: Chi-square = 42.14; df = 2; $p < 0.001$; IPM: Chi-square = 57.50; df = 2; $p < 0.001$), *E. flabellatus* (ORG: Chi-square = 44.13; df = 2; $p < 0.001$; IPM: Chi-square = 71.49; df = 2; $p < 0.001$).

In the IPM olive grove, a greater number of captures were registered, for all the species, in those plots treated with dimethoate, in which the values were significantly higher than those registered in the control plots ($p < 0.05$). On the contrary, those plots treated with pyrethrins presented the minimum capture values, being generally significantly lower than those observed in the control plots ($p < 0.05$).

In the organic olive grove, the beneficial insect species presented higher post-treatment capture rates in the control plots, where the values were significantly higher than those observed for the plots treated with either of the two insecticides under study.

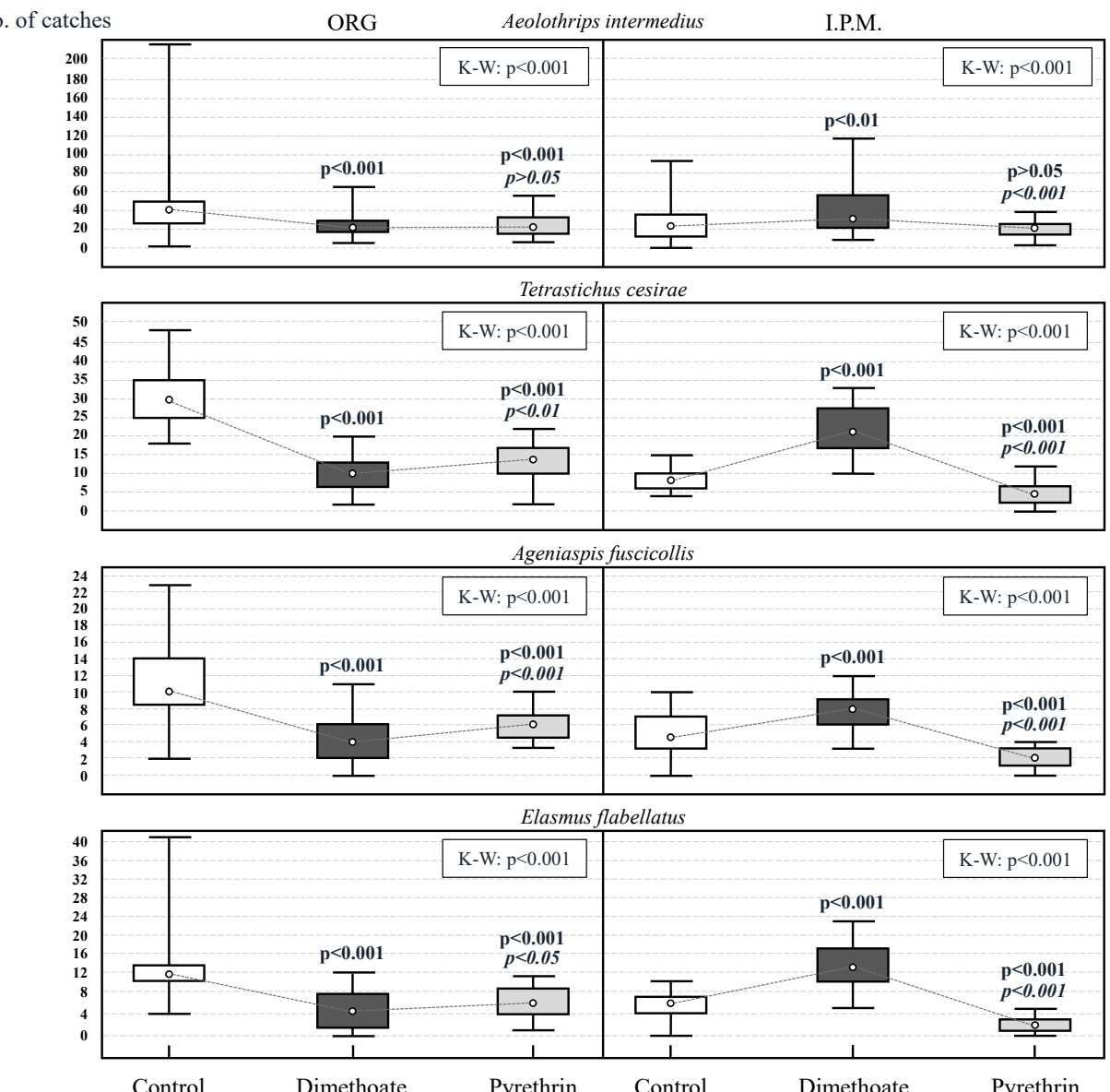

**Figure 5.** Statistic capture values (median, quartile 25, quartile 75, maximum and minimum) of the main beneficial species in the control plots (white color), treated with organophosphate Dimethoate 40% © (dark color) and pyrethrins (light color), of the two olive groves (organic/IPM). The *p*-value (Kruskal–Wallis test) is indicated in the upper right corner. The *p*-values indicated above the boxplots correspond to the comparison with respect to the control plot (**bold letters**) and between the two insecticides (*italic letters*).

## 4. Discussion

In the first part of this study, which corresponds to the data from the control plots, which during the study remained free of any insecticide action. The capture data of the plots reflect the long-term effect that the use of each of the two compared insecticides is having on the relative abundance of the selected species. The greater abundance of beneficial species in the organic olive grove, compared to the IPM olive grove, agrees with the results of studies carried out in olive groves in Spain [33,86] and Portugal [87]. Given that the main difference between the types of management (IPM and organic) mainly affects the insecticide used and the application frequency, the results are consistent with those reported by Santos et al. [87], since the lethality produced by dimethoate on beneficial fauna is the cause of an important alteration in the agroecosystem balance in IPM olive

groves [88,89]. This effect, together with the absence of dimethoate selectivity, has been indicated even for concentrations lower than the $LD_{50}$ [90], causing significant alterations in the reproductive process of surviving individuals [91], which significantly reduces their success rate. In this sense, Pascual et al. [92] and Nikolova et al. [80] point out that the reduction in the abundance of natural enemies is an obvious consequence of the regular application of synthetic pesticides in agroecosystems. This dependence on synthetic insecticide applications for pest control negatively affects the predatory efficacy of natural enemies, such as the lacewing larvae [33]. In contrast, the higher population of natural enemies in organic olive groves results in higher predatory activity [33].

As presented in our results on IPM management, which is regularly treated with dimethoate, the experimental application of this insecticide triggers an increase in the number of captures in all the selected natural enemy species in relation to the control plots. As indicated before, among the sublethal effects of insecticides, several alterations in behavior are mentioned, such as agitation, hyperreflexia, irritability, and repellency [3–9]. Affected individuals have a tendency to move towards the insecticide-free surfaces, which explains a higher rate of capture in the sticky traps of the treated plots, so these results are in line with what has been previously observed [18,19,33]. Therefore, in the IPM olive grove, the increase in post-treatment capture values reflects the existence of a repellent effect, which leads insects to areas free of the pesticide. This is a characteristic effect of populations submitted to selection pressure, which favors lineages better adapted to avoid the insecticidal action. This implies an increase in the capture rate in plots usually treated with this insecticide. Several authors have pointed out the acquisition of behavioral resistance, as a behavior modification, developed by populations frequently exposed to sublethal doses [3,5,6,17,26]. It is important to emphasize that the increase in the number of individuals in the traps in the treated areas can be misleading when interpreting it as a result of an increase in abundance in the olive groves after the application of dimethoate. Obviously, it would be an erroneous statement, since it would be due to the reaction induced in the behavior of the individuals affected by sublethal doses, and their reaction of fleeing towards surfaces free of the insecticide, as a survival mechanism, which is very different from a reaction due to its chromatic attraction. Repeated exposure to sublethal doses is therefore a necessary condition for the gradual acquisition of behavioral resistance and explains the absence of increased capture rates in organically managed plots experimentally treated with dimethoate since insects lack this capacity. Regarding the real insecticide dose absorbed by insects, it is impossible to determine the proportion of them that receive sublethal doses [93]. In certain species of insects, and particularly in species of medical importance, such as malaria vectors, the existence of "behavioral resistance" has been indicated as a result of repeated exposure to sublethal insecticide concentrations. However, authors such as Chareonviriyaphap et al. [17] have proposed the term "behavioral avoidance", which, unlike resistance, is the result of an innate, natural, and involuntary response or capacity. They proposed that, unlike behavioral resistance, behavioral avoidance can play an important role in reducing selection pressure, thus slowing the emergence and spread of physiological resistance. However, in this study, it could be concluded that the effect is related to acquired behavioral resistance since in the organic olive grove, where dimethoate had not been used prior to this experiment, insects lack a repellent reaction, which represents the main difference with respect to the insect populations in the IPM olive grove.

In both types of management (IPM and organic), plots experimentally treated with pyrethrins showed a decrease in the post-treatment capture rate compared to the control plots. On the one hand, in the organic olive grove, the absence of repellency of beneficial insects against pyrethrins is consistent with the scarcity of reports on resistance to this insecticide in the field [94]. On the other hand, in the IPM olive grove, experimental pyrethrin application corresponds to a reduction in the capture rate, an opposite effect to the dimethoate application, which makes it possible to rule out the possibility of cross-resistance of insects against both insecticides. A plausible explanation for the lack of repellency with respect to pyrethrins is probably attributed to their large knockdown effect [38,39,95], which involves rapid movement paralysis and eventual death [38]. In

addition to this effect, the rapid waste degradation and its short persistence, of a few days or only a few hours [39,44,96,97], imply a specific action in relation to that of conventional insecticides; therefore, under these conditions, the population of affected insects must be much lower, which leads to conclude that the selective pressure generated is greatly mitigated and practically imperceptible. This may explain why most of the literature on acquired resistance to pyrethrins corresponds to studies on their application in the control of pests of stored agricultural products [53–57], where storage conditions considerably limit the action of environmental factors that degrade pyrethrins.

## 5. Conclusions

The application of the induction-monitoring technique for the detection of behaviorally resistant lineages in olive groves under IPM and organic pest management, through the experimental application of pyrethrins and dimethoate as inducing insecticides, allowed for the detection of the existence of lineages in the IPM olive grove resistant to dimethoate, although not to pyrethrin. In the organic olive grove, where the product most widely used is pyrethrin, the relative abundance of beneficial insects was notably higher. Thus, no repellency reaction was detected in relation to both insecticides under study.

**Author Contributions:** Conceptualization, J.A.G.-G. and R.G.-R.; methodology, R.G.-R.; validation, J.A.G.-G. and M.S.-P.; formal analysis, J.A.G.-G.; investigation, J.A.G.-G.; resources, J.A.G.-G.; data curation, J.A.G.-G. and R.G.-R.; writing—original draft preparation, J.A.G.-G., R.G.-R. and M.S.-P.; writing—review and editing, J.A.G.-G. and R.G.-R.; visualization, R.G.-R.; supervision, R.G.-R.; project administration, R.G.-R. All authors have read and agreed to the published version of the manuscript.

**Funding:** This research received no external funding.

**Institutional Review Board Statement:** Not applicable.

**Informed Consent Statement:** Not applicable.

**Data Availability Statement:** Not applicable.

**Conflicts of Interest:** The authors declare no conflict of interest.

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
