# Peer review of "Monitoring and Inference of Behavioral Resistance in Beneficial Insects to Insecticides in Two Pest Control Systems: IPM and Organic"

_agronomy, doi:10.3390/agronomy12020538_

Round 1

Reviewer 1 Report

Dear Editor and Authors,

Guzman et al. made significant changes to the manuscript that improved its readability and clarity. The shortening of the title was appropriate, as it provides with fewer words an exact idea of the content of the manuscript.

Regarding Figure 1, it is necessary to cite the source (origin of the image).

Statistical procedures were better described in this version of the manuscript. Comparison of capture data by the Chi-square test is adequate.

The article is classified as Short Communication, although the number of experiments performed and its significance allows classifying it as a Full Research Paper.

Author Response

Point 1: Guzman et al. made significant changes to the manuscript that improved its readability and clarity. The shortening of the title was appropriate, as it provides with fewer words an exact idea of the content of the manuscript.

Regarding Figure 1, it is necessary to cite the source (origin of the image).

Statistical procedures were better described in this version of the manuscript. Comparison of capture data by the Chi-square test is adequate.

The article is classified as Short Communication, although the number of experiments performed and its significance allows classifying it as a Full Research Paper.

Response 1: Dear reviewer, thank you very much for your comments, we value them very positively. These comments and the previous ones (previous version) have helped us a lot in improving our work, for which we are very grateful.

In relation to Figure 1, the source of the image (both of this figure and of Figure 2) has been added.

Reviewer 2 Report

The manuscript is aimed to evaluate the response of natural enemies in olive groves under IPM and organic pest management after specific treatment. The study is well designed, clearly described, and scientifically interesting.

What about the semiochemicals produced by insects for example sex or aggregation pheromone produced by trapped insects. Moreover, repellent volatile organic compounds (VOCs) come from different weed flowers as you apply the experiments in spring, moreover, the yellow traps (as many natural enemies can attract to this color) have a role in this captured?  

Some minor comments are reported.

Line 6-7 could you please indicate the full address (City, county).

49 54 not clear.. Rewrite.

95 add the descriptor for all scientific names… check along with the text.

169 Both olive groves (Organic and IPM) were ……

171 – 175 here is not clear.. Did you apply Dimethoate also in the organic grove?.. Please re-write.

190-191 did you apply the trap before the experiments also? In order to have a look if there is some attraction toward the traps.

191-192 is 16 traps per plot? and in each block, you should have 48 traps?

Author Response

Point 1: The manuscript is aimed to evaluate the response of natural enemies in olive groves under IPM and organic pest management after specific treatment. The study is well designed, clearly described, and scientifically interesting.

Response 1: Dear reviewer, thank you very much for your comments, we value them very positively. These comments and the previous ones (previous version) have helped us a lot in improving our work, for which we are very grateful.

Point 2: What about the semiochemicals produced by insects for example sex or aggregation pheromone produced by trapped insects. Moreover, repellent volatile organic compounds (VOCs) come from different weed flowers as you apply the experiments in spring, moreover, the yellow traps (as many natural enemies can attract to this color) have a role in this captured? 

Response 2: Certainly, the insects captured in the traps can release (at least in the initial stages of their capture) some type of semiochemical with activity in individuals of the same species, which would probably affect in some way the arrival of successive individuals, and therefore to the total catches. However, be that as it may, this effect would occur equally in olive groves, regardless of whether or not the experimental treatment under study was applied to them. For this reason, we have believed that since it is a constant element, we have not taken it into account when comparing the figures between treated and control plots.

Point 3: Line 6-7 could you please indicate the full address (City, county).

Response 3: Information added in the new version of the manuscript.

Point 4: 49 54 not clear. Rewrite.

Response 4: This part of the text has been slightly modified. We hope we have fixed the problem in this new version.

Point 5: 95 add the descriptor for all scientific names… check along with the text.

Response 5: Information added in the new version of the manuscript.

Point 6: 169 Both olive groves (Organic and IPM) were ……

Response 6: Text changed in the new version.

Point 7: 171 – 175 here is not clear. Did you apply Dimethoate also in the organic grove? Please rewrite.

Response 7: In this new version of the manuscript, more information has been added to this part of the text to make it more understandable. We hope we have solved the problem.

Point 8: 190-191 did you apply the trap before the experiments also? In order to have a look if there is some attraction toward the traps.

Response 8: Indeed, thank you. As presented in the Material and Methods, the experimental insecticide treatment has been applied just before the installation of the traps. It is an interesting observation, since it is possible that the installation of the traps prior to the insecticide could affect the results, in the aspect that being impregnated by the insecticide could be repellent for the arrival of insects. This aspect will be studied in successive field studies. Thanks

Point 9: 191-192 is 16 traps per plot? and in each block, you should have 48 traps?

Response 9: Exactly, in each sample, the total number of traps placed in each plot was 16. However, in order to avoid the lack of randomization of the observed samples and prevent pseudoreplication effects, 6 traps of each plot were randomly selected to be observed.

Reviewer 3 Report

Your manuscript provides interesting results on the effects of pyrethrins. However, the manuscript raises various key questions. The manuscript has typos and was not carefully prepared. Manuscripts should be proofread by native English speakers for clarity and readability. The abstract does not represent the study well and should be rewritten. The abstract should include the purpose of the research, the main findings, and one or two concluding sentences. The introduction section contains repeated sentences and should be more concise. Some statistical methods were insufficient to interpret the results obtained in some experiments (abundance species estimators should be used to assess abundance). Furthermore, it should be clarified that statistical methods must have a logical order.
1. A summary is required in the Introduction, Materials and methods, Results and Discussion sections.
2. Some informative statistical methods (X2, t- or F-value, degrees of freedom and p-value) are missing from the results. For each experiment, the data should be statistically analyzed and included in the results section.
3. For other modifications, please view the attached PDF.

Author Response

Dear reviewer, thank you very much for sending us your comments. We believe that the manuscript has been significantly improved. However, please bear in mind that given the limited timeframe that the Agronomy editor has allowed us for this last stage of corrections (5 days from receipt of your comments), some of the suggested changes could not be addressed. In large part also because its realization implies a conflict with the changes suggested by the other three referees, and which were made in an earlier stage of corrections. However, most of your comments have been addressed to improve the manuscript. For greater clarity, it would have been preferable if the suggestions had been sent in English language, because the short time frame has not allowed us to have official Chinese-Spanish translators, which has made it significantly difficult for us. We hope you find it suitable for publication in Agronomy.

Here we explain the changes made:

Point 1: Manuscripts should be proofread by native English speakers for clarity and readability.

Response 1: Dear reviewer, the manuscript has been firstly translated by a professional English translator and subsequently reviewed by MDPI's English editing services, which have verified the correct use of grammar and common technical terms. MDPI uses experienced English speaking editors, so we believe that the level of English and grammar has been sufficiently reviewed. Finally, the rest of the reviewers consider that the grammatical level of English and writing is correct.

Point 2 (& point 4): The abstract does not represent the study well and should be rewritten. The abstract should include the purpose of the research, the main findings, and one or two concluding sentences.

Response 2: We agree with your indications, although we have not been allowed to provide more information, due to the limitation of only 200 words for the abstract, at most. The abstract of the manuscript has already been published on the journal's website, as a planned paper, therefore it has been reviewed by the editors and they have suggested the current form.

Point 3: The introduction section contains repeated sentences and should be more concise.

Response 3: A modification of the introduction has been made, taking into account your comments on the PDF, as well as the comments of the other reviewers.

Point 5: Some informative statistical methods (X2, t- or F-value, degrees of freedom and p-value) are missing from the results.

Response 5: Along with the results of the statistical analyses, other important values have been added, such as the degrees of freedom (df).

Point 6: For other modifications, please view the attached PDF.

Response 6: In the new version of the manuscript, all your annotations in the PDF have been considered. Therefore, we hope that this new version has solved the problems that you mentioned here. However, certain changes that you comment on in the PDF, such as the deletion of Table 1, have not been made as this conflicted with the other reviewers' comments. Table 1 was expressly requested by another reviewer to report the details of the application of these products in each olive grove (Active ingredient per ha; date of application etc).

Reviewer 4 Report

This MS is generally well-written and scientifically sound,  the minor editorial comments in the body of the attach original MS.

Author Response

Point 1: This MS is generally well-written and scientifically sound, the minor editorial comments in the body of the attach original MS.

Response 1: Dear reviewer, thank you very much for your comments, we value them very positively. These comments and the previous ones (previous version) have helped us a lot in improving our work, for which we are very grateful.

Point 2: Chlorpyrifos 48% ©: This is a very toxic organophosphate. As you only make reference to Dimethoate, was Chlorpyrifos used in this study?

Response 2: In relation to this comment on the PDF, it should be noted that we have not used this insecticide in this research. The mention of this organophosphate in the text is only due to the fact that it is quite usual in the control of the branches-borer moth, Euzophera pinguis by the farmers.

Point 3: “Repeated exposure to sublethal doses is therefore a necessary condition for the gradual acquisition of behavioral resistance, and explains the absence of increased capture rates in organically managed plots experimentally treated with dimethoate, since insects lack this capacity.” I wonder if insects do not move between treatments?

Response 3: We agree that the application of dimethoate, by inducing repellency, can facilitate the evacuation of part of the affected insects to areas free of the insecticide. This would be more evident in the marginal areas, especially in the contact areas between IPM and organic Olivares. In fact, determining this migration constitutes the objective of our next work.

Round 2

Reviewer 3 Report

The authors have incorporated all suggestions and comments into the revised version, now the manuscript seems much clear. There are some minor points to be corrected:

Introduction section: change “Hem.” by “Hemiptera”

Discussion section (second paragraph): Sentence starting “As presented In our results on IPM implementation, which is…

Author Response

Point 1: The authors have incorporated all suggestions and comments into the revised version, now the manuscript seems much clear. There are some minor points to be corrected:

Introduction section: change “Hem.” by “Hemiptera”

Discussion section (second paragraph): Sentence starting “As presented in our results on IPM implementation, which is…

Response 1: Dear reviewer, thank you very much for your comments, we value them very positively. These comments and the previous ones (previous version) have helped us a lot in improving our work, for which we are very grateful.

In relation to the comment of the introduction, we have replaced "Hem.", with "Hempitera", as you have suggested.

In relation to the comment of the discussion, we have replaced the first sentence of the second paragraph with the one you have suggested.

We hope you find the new version of the manuscript suitable for publication in Agronomy.

This manuscript is a resubmission of an earlier submission. The following is a list of the peer review reports and author responses from that submission.

Round 1

Reviewer 1 Report

Title: “Field Experiment for the Inference of Behavioral Resistance in Beneficial Insects to Insecticides Used in Two Olive Groves with Different Pest Control Systems: Dimethoate (IPM) and Pyrethrins (Organic Management)

Authors: Gomez - Guzman et al.

It is an manuscript that provides data, some novel, for the effects of two insecticides on beneficial insects of olive pest. The results contribute to the growing body of evidence that some insecticides have sublethal effects.

In general, the manuscript is not well written and the experiments not well performed. The results may have also a practical interest, although the authors could highlight/discuss more the practical implementation of their findings. I detect major methodological flaws and important problems regarding the clarity of, and correspondence between, the results and the conclusions, what raised doubts about whether the scientific contribution is sound enough to merit publication in Agronomy. However, the author should take into account some comments that are listed below.

Keywords should be in alphabetic order. Also, keywords serve to widen the opportunity to be retrieved from a database. To put words that already are into title and abstracts makes KW not useful. Please choose terms that are neither in the title nor in abstract.

Introduction

The manuscript does lack of an exhaustive introductory part. Moreover, whether you focus only on Spain you depict a very narrow scenario, I would suggest to give to the manuscript a "broader view" instead of a local one. This section need to be up to date concerning the recent findings within the investigated topic in a more comprehensive IPM or biological framework.

P1. This implies …. lethal dose (LD50). Incorrect statement. A sublethal dose/concentration is defined as inducing no apparent mortality in the experimental population. Please rewrite.

P2L2-3 Please add some examples of natural enemies

P3 change ‘humans make pyrethrins more ecologically’ to ‘humans make natural pyrethrins more ecologically’. Please rewrite

-Last paragraph of Introduction. Please rephrased and then the authors should clearly state the hypothesis they want to test and how they solved.

M&M

Material & methods could be enhanced by a careful description and by rearranging logically the experimental procedures. Please rewrite the M&M

First paragraph, the authors should mention ecological management to introduction.

P4L3 add order and family after pest scientific name

P5L1 4% pyrethrins… the authors mean natural or synthetic pyrethrins

- Provide the insecticide details in a new topic. give a.i per hectare of each insecticide

- Provide details about the methodology used.

- What about the other parts of the two olives fields (eco and ipm). Farmers spray for any pest etc?

Why the authors count only one plot?

Why the authors transformed they data? The authors should not use transformed data on figures.

Results

-Change entomophagous to natural enemies

-Please provide identified species for each category (IPM – Organic) in different line or table.

Fig 3. Please give two figures, side by side, one for IPM and one for Organic.

-In the IPM olive grove, all species had higher capture rates in those plots treated with dimethoate’, the authors should explain what it means capture rates, it’s the same with capturing means? Or capture values?

Discussion

The discussion have a lack of information about the sublethal effects of insecticides on beneficial insects, and in my opinion the probable effects on beneficial insects was poorly explored.

Reviewer 2 Report

The study is interesting, but there are two major flaws. The first is that there is only one season worth of data. The more serious flaw is that the the experiment was set up with three blocks of each treatment per farm but traps from only one block were used on each sampling date. The traps are pseudoreplicates; the true replicates are the blocks. Therefore, the statistical analysis is invalid. A subsample of traps from each block needs to be used on each sampling date. This is the primary reason why I am rejecting the manuscript. Finally, the lack of randomization is an issue and needs to be justified. The possible effects of this lack of randomization need to be included in the discussion as well.